# BEHAVIOR-INFUSED EVIDENCE-FIRST REASONING: BRIDGING THE OFFLINE-ONLINE GAP IN RECOMMENDATION

## ABSTRACT

Modern recommender systems often exhibit an offline-online performance gap. A major reason for this is missing feedback: offline logged data lack feedback for items that are never shown by the production system, making it difficult to evaluate counterfactual outcomes. Large language models (LLMs), with their broad knowledge and reasoning capabilities, are promising backbones for reward models that impute this missing feedback. Given a user's interaction history and a candidate item, these models can judge whether a recommendation is a good fit. However, a vanilla LLM bases its judgment almost entirely on semantic information, ignoring behavioral signals and offering no justification for its assigned rewards. To overcome these limitations, we propose BRIEF (behavior-infused evidence-first reasoning), which constrains LLM generation to provide structured evidence before assigning rewards, and injects behavioral signals through adaptive logit biasing guided by a collaborative filtering (CF) model. Using online A/B test results from a mainstream video streaming platform, we show that offline evaluations from BRIEF correlate strongly with online business metrics. We also validate BRIEF's ability to synthesize high-quality rewards: using them for training-data augmentation improves downstream recommender performance, and its judgments show strong correlation with real user ratings.

## 1 INTRODUCTION

Reliable evaluation is vital for developing recommender system algorithms (Castells & Moffat, 2022). While online A/B tests are the most effective assessment, they are costly and inefficient (Wang et al., 2023). As a result, offline evaluation remains the primary validation approach before moving to online experiments, using metrics such as precision and hit rate (Herlocker et al., 2004; Hidasi & Czapp, 2023). Yet offline metrics can suffer from the offline-online gap existing in implicit-feedback recommender systems and fail to predict online performance (Krauth et al., 2020). This gap largely stems from exposure bias: we observe very little to no feedback from an item if the production system rarely or never surfaces it (Jeunen, 2019); a well-known subtype is popularity bias, where long-tail items are largely underrepresented (Bellogín et al., 2017).

A widely used approach to narrow the gap is off-policy evaluation (OPE): inverse propensity scoring (IPS) re-weights logged interactions to provide an unbiased estimate of the reward (Narita et al., 2021), but cannot handle unseen actions and is sensitive to inaccurate propensity estimates (Felicioni et al., 2022; Zhang et al., 2023); Direct Methods (DM) train a reward model directly to predict outcomes, inherit biases from the logged data, and struggle on out-of-distribution items. As large language models (LLMs) possess broad knowledge and strong reasoning abilities (Wang et al., 2024a), they are natural candidates for building reward models (see Figure 1(a)) that, given a user's item-interaction history and a candidate recommendation, can generate synthetic feedback (Zhang et al., 2024a).

Nevertheless, a vanilla application of LLMs as reward models is insufficient: *LLMs have no knowledge of task-specific user–item interaction patterns, and they cannot justify assigned rewards with evidence.* These limitations are shared by both zero-/few-shot prompting and advanced agentic user simulators (Zhang et al., 2024a; Bougie & Watanabe, 2025). As shown in Figure 1(b), collaborative signals are vital to recommendation, and LLMs excelling at semantic understanding but ignoring

Figure 1: **(a)** A reward model generates synthetic feedback for candidate movies based on the user's viewing history. **(b)** A recommendation can be relevant even with low semantic similarity to the user's interaction history when a strong collaborative pattern—users with similar histories also like the item—supports it. **(c)** A cross-domain recommender pushes the movie *Harry Potter* to a user who has shown strong franchise interest by reading multiple *Harry Potter* books, so the recommendation is likely relevant; yet it is labeled negative in offline evaluation because the user has never been exposed to the film and it is therefore absent from the ground-truth watch list.

behavioral information make unreliable judgments (Zheng et al., 2024; Hong et al., 2025; Wang et al., 2025b). Moreover, grounding each reward in specific evidence from the user's history makes the judgment process transparent and debuggable. Therefore, we propose BRIEF, enabling LLMs to provide high-quality recommendation evaluation through behavior-infused evidence-first reasoning. Specifically, it enforces evidence-first constrained generation—producing structured evidence before assigning rewards—which avoids post-hoc rationalization and creates a single, predictable control point for injecting behavioral signals by adaptively adjusting output token logits using a collaborative filtering (CF) model. Unlike methods that integrate collaborative signals by training LLMs with special item tokenization (Rajput et al., 2023) or through multi-round conversations, BRIEF operates at the decoding stage and is thus training-free. Consequently, BRIEF is a lightweight approach that integrates behavioral information and justifies each assigned reward.

We assess BRIEF's ability to narrow the offline-online gap by comparing its offline scores with business metrics from online A/B tests on a video streaming platform, measuring how well these scores predict online performance. Because such offline-online correspondence data are rare, we conduct additional experiments on public and production datasets: we augment training data with BRIEF's synthetic feedback to enhance recommender models, and we show that BRIEF's judgments correlate strongly with user ratings, outperforming baselines. We further validate BRIEF in cross-domain recommendation, where the offline–online gap is especially wide because exploratory signals that bridge domains are underrepresented in logged data (Figure 1(c)) (Chen et al., 2021; Xu et al., 2024). We summarize our contributions as follows:

- We introduce BRIEF, a lightweight method that enables LLMs to produce evidence-justified recommendation evaluations and fuses behavioral with semantic signals via adaptive logit biasing. To the best of our knowledge, this is the first to infuse collaborative signals at decoding time.

- We demonstrate empirically that BRIEF delivers the most reliable offline evaluation of recommender models, with additional validation in cross-domain recommendation, where the offline-online gap is especially large.

- We show that BRIEF directly improves recommendation quality by imputing missing feedback in training data, and that it produces evaluations that agree with real user ratings.

## 2 METHODOLOGY

### 2.1 PROBLEM FORMULATION

Let $\mathcal{I} = \{i_1, \ldots, i_n\}$ denote the universe of items. For a given user, let the interaction history be the set $H_u \subseteq \mathcal{I}$. Given this history and a recommended candidate item $i_j \in \mathcal{I} \setminus H_u$, our objective is to

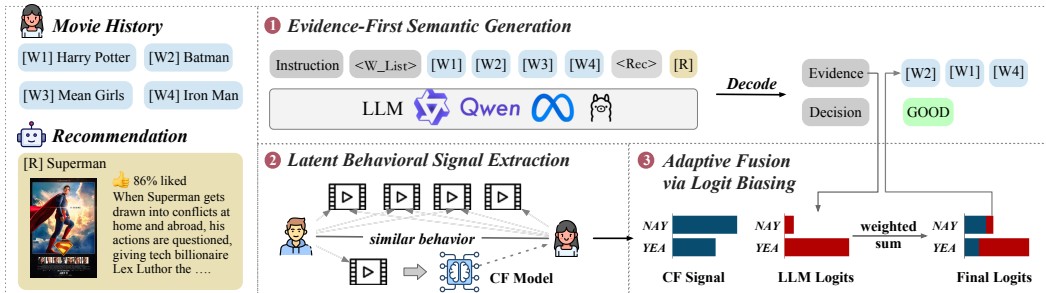

Figure 2: Overview of BRIEF. **(1) Evidence-First Semantic Generation**, where the LLM provides structured evidence prior to assigning rewards; **(2) Latent Behavioral Signal Extraction**, which computes behavioral intensities using a CF model; and **(3) Adaptive Fusion via Logit Biasing**, which adaptively adjusts token logits with behavioral signals.

develop a reward model $\mathcal{R}$ that generates a binary reward signal $r \in \{0, 1\}$, where $r = 1$ denotes a positive (relevant) recommendation and $r = 0$ denotes a negative one. The generated rewards can be used to evaluate recommender models offline, thereby helping to narrow the offline-online gap.

Our proposed method, BRIEF, consists of three modules—Evidence-First Semantic Generation, Latent Behavioral Signal Extraction, and Adaptive Fusion via Logit Biasing. An overview of BRIEF is presented in Figure 2. We introduce each module in turn.

## 2.2 EVIDENCE-FIRST SEMANTIC GENERATION

Large language models, with extensive world knowledge and reasoning ability, can judge the relevance of a recommendation through semantic understanding, either from their own knowledge or with the help of item metadata (e.g., genre, synopsis). This judgment is inherently based on identifying relationships between a candidate item and a user's interaction history, such as items belonging to the same franchise, sharing a genre, or serving as complementary products. Because this reasoning is tied to specific past items, a trustworthy positive reward must be explicitly supported by citing a subset of history items as verifiable evidence. Therefore, BRIEF enforces an evidence-gathering step—**the LLM identifies "evidence items"** $E_u \subseteq H_u$ **if and only if it generates** $r = 1$—reformulating the task from a simple binary classification into a structured generation problem.

However, simply generating evidence after making the relevance decision encourages post-hoc rationalization, where the model invents plausible but unfaithful explanations to fit its judgment. Therefore, BRIEF constrains the LLM to commit to providing evidence before it can assign a positive reward. This guides the model to follow a more grounded reasoning path where justification becomes a precondition for the judgment, not an afterthought. Nevertheless, a candidate item may be highly relevant when it exhibits strong behavioral correlation with a user's past items despite low semantic similarity. Consequently, evidence-first reasoning alone is insufficient.

> **Enforcing Evidence-First Reasoning**
>
> Must list evidence title(s) if you decide the recommendation is **relevant**; must provide none if you decide it is **irrelevant**.
> Return only this JSON object:
> ```
> {
>   "evidence": ["<title-1>", "..."],
>   "is_relevant": "<YES|NO>"
> }
> ```

## 2.3 LATENT BEHAVIORAL SIGNAL EXTRACTION

From a behavioral standpoint, a recommendation is a good fit when users with similar tastes have consumed the candidate item. Such similarity of taste is inferred from user interaction histories, which reveal latent item co-consumption patterns. Thus, judging relevance requires behavior-driven similarity scores. This principle underlies modern retrieval-stage recommenders, which embed items, learn user representations, and retrieve nearest neighbors. Below, we outline two ways to derive latent behavioral signals:

**User-Item Relevance Scoring** A straightforward way is to adopt the standard inference pipeline of modern sequential recommenders (e.g., GRU4Rec (Hidasi et al., 2015), SASRec (Kang & McAuley, 2018)). These models are trained for next-item prediction by processing a user's sequence of historical interactions to learn a dynamic user representation vector. A relevance score for a candidate item is then computed by taking the dot product of this final user representation and the candidate's item embedding. This resulting score can be directly used as the behavioral signal.

**Pairwise Item-Item Similarity** We can also derive a signal directly from the item embeddings learned by these recommenders. The item embeddings encode rich behavioral patterns, including co-consumption and higher-order relationships. We can compute the pairwise similarity between a candidate item $j$ and each item $i$ in a user's history $H_u$ as $\text{sim}(i, j) = \cos(\hat{\mathbf{z}}_i, \hat{\mathbf{z}}_j)$ on $\ell_2$-normalized embeddings $\hat{\mathbf{z}}$. The behavioral signal is then calculated as the average of the top-$k$ similarity scores.

Both of these methods yield **a collaborative filtering (CF) reward model that, given a user $u$ and a candidate item $i$, produces** $p(r = 1|u, i)$. However, because these CF models are trained on logged interactions, they inherently suffer from exposure bias. This limits their ability to generalize to out-of-distribution items and makes them less effective at bridging the offline-online gap. In contrast, since LLMs are not trained on these logs, their judgments via zero-shot prompting are not driven by the same exposure bias, making them particularly beneficial for cold-start items.

### 2.4 ADAPTIVE FUSION VIA LOGIT BIASING

Because BRIEF constrains the LLM to provide evidence if and only if it assigns a positive reward, the model's generative trajectory irrevocably splits between a positive and negative outcome immediately after emitting the `"evidence":` trigger: the model must choose between an empty list (a negative judgment) or beginning a non-empty list (a positive judgment). This process creates **a single, predictable control point during decoding** where behavioral signals can be infused to steer the LLM's semantic reasoning.

At this control point, we treat the token `[` as the YEA token (committing to a positive reward) and `[]` as the NAY token. After emitting the `"evidence":` trigger, BRIEF linearly scales the behavior signal computed in Section 2.3 to a single intensity score $\sigma \in [-1, 1]$, where a positive value indicates collaborative support for recommending $i$ to $u$. Crucially, the intervention strength adapts to the LLM's own semantic confidence. A skewed probability distribution over the YEA and NAY logits implies high confidence that should be respected, while a flat distribution implies uncertainty where external guidance from behavioral patterns is most useful. We quantify this by computing the entropy over the YEA and NAY logits and normalizing it to $E_n \in [0, 1]$, where higher $E_n$ means greater uncertainty. The final logit bias is

$$\Delta = \beta \cdot (1 + E_n) \cdot \sigma,$$

with hyperparameter $\beta$ controlling the base strength. At this decoding step, BRIEF applies a push-pull update by adding $\Delta$ to the YEA logit and subtracting $\Delta$ from the NAY logit. Through subsequent softmax and sampling, this (i) reinforces the LLM's original decision when behavior and semantics agree, (ii) flips the decision when strong behavioral evidence opposes a low-confidence semantic judgment, or (iii) leaves the decision unchanged when the LLM is highly confident or $\sigma$ is too weak.

## 3 EXPERIMENTS

We measure how well BRIEF's offline scores correlate with online business metrics, using A/B test results from a major video streaming platform. Because such offline-online correspondence data are rare, we supplement this analysis with proxy downstream tasks on both public and internal industrial datasets to further validate the quality of BRIEF's synthetic feedback. The details are provided below.

### 3.1 EXPERIMENTAL SETUP

**Datasets** We conduct experiments on large-scale internal industrial data and two public datasets from the Amazon Reviews 2023 collection (Hou et al., 2024): *Movies and TV* and *CDs and Vinyl*. While details of the industrial dataset cannot be disclosed, statistics for the public datasets after pre-processing are summarized and provided in Table 1.

Table 2: Offline-online gap (MAE, percentage points; lower is better) across offline evaluation methods. *Normal*: using logged ground-truth interactions only. *SASRec*: behavior-only imputation. *Zero-shot*: semantic scoring without behavioral signals. *Non-adaptive*: BRIEF without entropy-based adaptive fusion. Best per column in **bold**, second best underlined. P denotes *precision*.

| Method | P@10 | P@20 | HitRate@10 | HitRate@20 | nDCG@10 | nDCG@20 | MAP@10 | MAP@20 | MRR |
|--------|------|------|-----------|-----------|---------|---------|--------|--------|-----|
| Normal | 2.9246 | 2.8206 | 3.1324 | 3.2415 | 2.0410 | 2.2734 | 1.1585 | 1.3330 | 1.0233 |
| SASRec | 2.8992 | 2.8055 | 3.1287 | 3.2565 | 2.0228 | 2.2607 | 1.1406 | 1.3165 | 0.9687 |
| Zero-shot | 2.8767 | 2.7443 | 3.1317 | 3.2300 | 2.0505 | 2.2682 | 1.1787 | 1.3418 | 1.0376 |
| Non-adaptive | 2.8965 | 2.7233 | 3.1353 | 3.2333 | 2.0613 | 2.2733 | 1.1947 | 1.3469 | 1.0485 |
| BRIEF | **2.6879** | **2.4490** | **3.1097** | **3.2105** | **2.0041** | **2.2289** | **1.1285** | **1.2934** | **0.9379** |

**Large language models** We use Qwen3-32B (Yang et al., 2025) as the backbone LLM for experiments on internal industrial data. For public data, BRIEF and all LLM-based baselines adopt Qwen3-14B to ensure a fair comparison. All experiments run on $4\times$ NVIDIA A100-SXM4-80GB GPUs,

Table 1: Statistics of pre-processed public datasets.

| Dataset | # Users | # Items | # Ratings |
|---------|---------|---------|-----------|
| CDs & Vinyl | 61,944 | 56,171 | 494,376 |
| Movies & TV | 303,310 | 71,543 | 2,254,458 |

with inference accelerated by vLLM (Kwon et al., 2023). For both models, we use a sampling-based decoding strategy with a temperature of $T = 0.7$, top-$p = 0.8$, and top-$k = 20$.

**Recommender models** We adopt SASRec (Kang & McAuley, 2018) to extract latent behavioral signals within BRIEF. SASRec is also evaluated as a reward model baseline in Sections 3.2 through 3.4.

**Advanced baselines** We rigorously evaluate BRIEF against several state-of-the-art (SOTA) LLM-based baselines. These include: a standard **zero-shot prompting** approach; **PUB** (Ma et al., 2025), which constructs detailed user personalities to prompt an LLM for behavior simulation; **Agent4Rec** (Zhang et al., 2024a), an LLM agent equipped with dedicated profile and memory modules to simulate user dynamics; and **RecAgent** (Wang et al., 2025a), which leverages flexible and efficient user profile modules to simulate user behaviors. Our implementation of PUB is from scratch, while our Agent4Rec and RecAgent models are adapted from their official codebases.

## 3.2 EFFECTIVENESS OF NARROWING THE OFFLINE-ONLINE GAP

We use online A/B test data collected from a mainstream video streaming platform and compare two recommender treatments, a control (A) and a variant (B). Standard offline evaluation ranks each model's top-$k$ recommendations per user and scores them against held-out interactions using *recall*, *precision*, *hit rate*, normalized discounted cumulative gain (*nDCG*), mean average precision (*MAP*), and mean reciprocal rank (*MRR*). In our setting, rather than relying only on future interactions, BRIEF evaluates every recommended item in the top-$k$ that does not match a future interaction; these additional rewards are then combined with ground-truth interactions to score the treatments. Because BRIEF assigns rewards only to items that appear in the top-$k$, the total number of relevant items per user is unknown, so *recall* cannot be computed in this setting.

Table 2 reports the offline-online gap for several offline evaluation methods. For each offline metric, we compute the *relative lift* of B over A offline as $100 \times (M_B - M_A)/M_A$ and compare it to the *online* relative lift measured by the business metric—the total number of streaming hours—computed as $100 \times (H_B - H_A)/H_A$. The gap for a metric is the mean absolute error (MAE) between these two lifts (lower is better); e.g., if offline reports $+5\%$ and online reports $+1\%$, the MAE is $4\%$. Values are averaged over three independent pairs of model treatments.

From Table 2, two patterns stand out. First, MAE is generally smaller for ranking metrics (*MAP*, *MRR*, *nDCG*) than for count-like metrics (*precision*, *hit rate*), suggesting that ranking-style offline metrics track online streaming hours more closely. Second, method-wise, vanilla zero-shot prompting reduces the offline-online gap by only $0.08\%$ on average—essentially negligible—whereas SASRec is a strong second, improving over normal evaluation by $1.18\%$ on average, underscoring the primacy of behavioral signals in recommendation. Notably, **BRIEF is consistently the best across all metrics and cutoffs**, reducing MAE relative to normal evaluation, SASRec, zero-shot prompting, and the non-adaptive ablation by $4.51\%$, $3.37\%$, $4.45\%$, and $4.84\%$, respectively. Overall, these

Table 3: Offline–online gap (MAE, percentage points; lower is better) across offline evaluation methods in the *cross-domain recommendation* setting. Best per column in **bold**, second best underlined.

| Method | P@10 | P@20 | HitRate@10 | HitRate@20 | nDCG@10 | nDCG@20 | MAP@10 | MAP@20 | MRR |
|---|---|---|---|---|---|---|---|---|---|
| Normal | 4.4748 | 2.8986 | 4.4250 | 3.0373 | 10.0293 | 8.4293 | 13.9234 | 13.0235 | 13.0750 |
| Zero-shot | 3.1076 | **2.0553** | **4.2400** | 3.0388 | 9.4839 | 8.0824 | 13.3285 | 12.5841 | 11.9340 |
| BRIEF | **2.9421** | 2.0578 | 4.3894 | **3.0216** | **9.2220** | **7.9369** | **12.7187** | **12.1501** | **11.7056** |

Table 4: MF performance after training on datasets augmented with different reward sources (higher is better). *Random Augmentation*: random positives; *SASRec*: behavior-only labels from collaborative signals; *Zero-shot*: semantic-only labels; *Non-adaptive*: BRIEF without entropy-based adaptive fusion. Best per column in **bold**, second best underlined. P denotes *precision* and R denotes *recall*.

| Method | R@5 | R@10 | R@15 | R@20 | P@5 | P@10 | P@15 | P@20 |
|---|---|---|---|---|---|---|---|---|
| No Aug. | 4.1013 | 6.7835 | 8.9380 | 10.8518 | 1.0755 | 0.9347 | 0.8390 | 0.7738 |
| Random Aug. | 4.0979 | 6.7788 | 8.9366 | 10.8562 | 1.0751 | 0.9340 | 0.8389 | 0.7740 |
| SASRec | 4.1158 | 6.8035 | 8.9566 | 10.8773 | 1.0839 | 0.9391 | 0.8422 | 0.7768 |
| Zero-shot | 4.1173 | 6.8024 | 8.9583 | 10.8691 | 1.0829 | 0.9388 | 0.8424 | 0.7766 |
| Non-adaptive | **4.1398** | 6.8210 | 8.9829 | 10.9019 | 1.0916 | 0.9425 | 0.8454 | 0.7796 |
| BRIEF | 4.1397 | **6.8237** | **8.9848** | **10.9041** | **1.0918** | **0.9430** | **0.8456** | **0.7797** |

results indicate that while semantic understanding helps judge relevance, fusing it with behavioral signals—and doing so adaptively—yields the closest alignment between offline and online outcomes.

**Cross-domain recommendation** We evaluate BRIEF in a cross-domain setting where the video recommendations also leverage users' book consumption histories, using one pair of model treatments. From Table 3, MAE values are markedly larger than in Table 2, confirming that the offline–online gap is especially wide in cross-domain recommendation. In this setting, ranking metrics deviate more from the online business metric (total streaming hours) than count-like metrics. **BRIEF remains consistently the best across all metrics and cutoffs**, increasing its average relative improvement over normal evaluation to $11.59\%$. Vanilla zero-shot prompting becomes highly competitive in this scenario—improving over normal evaluation by $9.97\%$ and trailing BRIEF by only $1.86\%$—highlighting the value of pure semantic signals that are not driven by the logged bias in capturing exploratory signals that bridge domains.

### 3.3 EFFECTIVENESS OF AUGMENTING RECOMMENDER TRAINING DATA

Because offline–online correspondence data (from A/B tests) are scarce and costly, we additionally assess reward quality via a proxy task: data augmentation for training recommenders. The premise is that a higher-quality reward model produces more useful positive labels; training on these labels should yield better downstream recommender performance.

**Internal industrial data** We use matrix factorization (MF) as the base recommender and compare five variants for generating additional positives. Table 4 reports results averaged over three independent days. Random augmentation slightly degrades performance on average ($-0.03\%$) yet shows small gains at *recall@20* and *precision@20*, suggesting mild noise can occasionally improve robustness. Single-signal augmentations—SASRec and vanilla LLM prompting—are nearly tied, improving over no augmentation by $0.39\%$ and $0.37\%$, respectively. The non-adaptive fusion further lifts the average margin to $0.79\%$. **BRIEF is best across almost all metrics and cutoffs**, with average improvements of $0.81\%$ over no augmentation, $0.42\%$ over SASRec, $0.44\%$ over zero-shot prompting, and $0.02\%$ over Non-adaptive. On this task, BRIEF's edge over non-adaptive fusion is trivial. However, since this task is evaluated on logged data, the results may suffer from exposure bias.

**Movies & TV dataset** We use SASRec as the base recommender and compare methods for generating additional positives. Augmenting the training data with synthetic positives generated by BRIEF leads to substantial gains in recommendation performance. As shown in Table 5, our method significantly outperforms the unaugmented baseline across all evaluation metrics, improving *nDCG@20* by

Table 5: Results for training data augmentation on **Movies & TV**. We evaluate using *recall* (R), *precision* (P), and *nDCG* at various cutoffs. Best per column in **bold**, second best underlined.

| | Movies & TV | | | | | |
| Model | R@10 | P@10 | nDCG@10 | R@20 | P@20 | nDCG@20 |
|---|---|---|---|---|---|---|
| No Aug. | 0.0263 | 0.0026 | 0.0091 | 0.0395 | 0.0020 | 0.0127 |
| Popularity | 0.0 | 0.0 | 0.0 | 0.0132 | 0.0007 | 0.0034 |
| Zero-shot | 0.0395 | 0.0039 | 0.0132 | 0.0395 | 0.0020 | 0.0132 |
| PUB | 0.0263 | 0.0026 | 0.0091 | 0.0395 | 0.0020 | 0.0121 |
| Agent4Rec | 0.0658 | 0.0066 | 0.0204 | 0.0921 | 0.0046 | 0.0270 |
| RecAgent | 0.0 | 0.0 | 0.0 | 0.0132 | 0.0007 | 0.0033 |
| BRIEF | **0.0714** | **0.0071** | **0.0224** | **0.1266** | **0.0063** | **0.0347** |
| item-item | 0.0395 | 0.0039 | 0.0132 | 0.0790 | 0.0039 | 0.0228 |
| non-adaptive | 0.0 | 0.0 | 0.0 | 0.0790 | 0.0039 | 0.0185 |
| no-behavior | 0.0526 | 0.0053 | 0.0157 | 0.0790 | 0.0039 | 0.0222 |

over $173\%$. Furthermore, BRIEF surpasses all competing augmentation strategies, with the strong agent-based model, Agent4Rec, being the next-best competitor.

The performance of the baseline methods highlights the difficulty of effective data augmentation. Naive strategies like adding popular items (Popularity) or even some advanced methods like RecAgent degrade the recommender's performance, demonstrating that the quality and relevance of the synthetic data are paramount. A standard zero-shot LLM provides a modest lift, suggesting that semantic understanding alone can uncover some useful signals, but it is not sufficient for achieving significant gains.

Our ablation studies confirm that BRIEF's effectiveness stems from its core design principles. The no-behavior variant, which relies solely on evidence-first semantic reasoning, still outperforms most baselines but is significantly weaker than the full BRIEF model. This result underscores the critical role of infusing behavioral signals. Moreover, the poor performance of the non-adaptive variant demonstrates that simply combining signals is insufficient; the adaptive fusion mechanism, which accounts for the LLM's confidence, is essential for generating high-quality training examples that boost, rather than hinder, performance.

### 3.4 CORRELATION BETWEEN USER RATINGS AND GENERATED REWARDS

To directly evaluate the quality of the synthetic rewards, we measure their correlation with explicit user ratings, which act as a ground-truth signal for user preference. We compute Spearman's $\rho$ and Kendall's $\tau$ rank correlation between the percentage of positive reward scores generated by each model and users' explicit ratings $(1-5)$. A higher correlation score signifies that a model's rewards more accurately reflect genuine user preferences.

The results, presented in Table 6, demonstrate that BRIEF consistently achieves the strongest alignment with user ratings across both datasets. It substantially outperforms the standalone collaborative filtering (SASRec) and vanilla LLM (Zero-shot) approaches. The near-zero correlation of the zero-shot baseline on the *Movies & TV* dataset highlights the insufficiency of relying on semantic signals alone. While advanced agentic baselines like RecAgent show strong performance, BRIEF matches or exceeds their ability to generate rewards that reflect user taste.

Our ablation studies validate the design of BRIEF. Removing the behavioral signal entirely causes a dramatic drop in performance, confirming that the fusion of collaborative patterns is essential. Furthermore, the non-adaptive fusion variant struggles on the complex *Movies & TV* dataset, indicating that the adaptive mechanism—which adjusts the behavioral signal based on the LLM's semantic confidence—is critical for robust performance. This confirms that BRIEF's ability to dynamically balance behavioral and semantic signals is key to its success.

### 3.5 EFFECT OF $\beta$

| Movie History | Recommendation | Zero-shot | BRIEF |
|---|---|---|---|
| A Haunted House, Terrifier 2, A Haunted House 2 | The Mean One | Decision: *Yes* | Decision: *Yes* ; Evidence: {A Haunted House, Terrifier 2, A Haunted House 2} |
| Blippi Wonders, Dinosaur Friends, Paw Patrol | Spiderman: Vexed by Venom | Decision: *No* | Decision: *No* |
| Ready Player One, Transformers, Bohemian Rhapsody | The Super Mario Bros Movie | Decision: *No* | Decision: *Yes* ; Evidence: {Ready Player One, Transformers} |

Figure 4: Zero-shot prompting vs. BRIEF: recommendation evaluation case studies.

We investigate the impact of the behavioral signal's base strength, controlled by the hyperparameter $\beta$, on the quality of the generated rewards. We vary $\beta$ from 5 to 40 and plot the Spearman's rank correlation with users' ratings for both public datasets in Figure 3.

The results show that the relationship between the performance of BRIEF and $\beta$ follows an inverted U-shaped curve. For both datasets, the optimal performance is achieved at $\beta = 25$. A value of $\beta$ that is too low provides an insufficient behavioral signal to steer the LLM's semantic-only reasoning, leading to poor performance. Conversely, a value that is too high allows the behavioral signal to overwhelm the LLM's judgment, causing the model to ignore valuable semantic information and leading to a sharp decline in reward quality. This analysis confirms that a carefully balanced fusion of behavioral and semantic signals is crucial for BRIEF.

Table 6: Correlation scores (Spearman's $\rho$ and Kendall's $\tau$). Best scores are **highlighted**; second-best are underlined.

| | Movies & TV | | CDs & Vinyl | |
|---|---|---|---|---|
| **Model** | Spear. | Kend. | Spear. | Kend. |
| SASRec | 0.6 | 0.4000 | 0.8000 | 0.6 |
| Zero-shot | -0.01 | 0.0 | 0.5643 | 0.3162 |
| PUB | 0.8207 | 0.7379 | 0.2236 | 0.1195 |
| RecAgent | 0.9 | 0.8 | 0.8 | 0.6 |
| Agent4Rec | 0.8 | 0.6 | 0.6 | 0.4 |
| BRIEF | **0.9000** | **0.8000** | **0.9000** | **0.8000** |
|   item-item | -0.3 | -0.2000 | 0.6 | 0.4000 |
|   non-adaptive | 0.1000 | 0.0 | **0.9000** | **0.8000** |
|   no-behavior | -0.3 | -0.2000 | 0.2000 | 0.2000 |

### 3.6 QUALITATIVE CASE STUDIES

To illustrate BRIEF's behavior-aware rewards and justifications, Figure 4 presents three case studies. We first note that vanilla zero-shot prompting with Qwen3-32B already yields reasonable judgments. *Case 1:* both vanilla prompting and BRIEF return YES; the cited evidence is consistent with the user's horror preferences—slasher/monster elements (Terrifier 2) and horror parody (A Haunted House)—matching the dark-comedy and over-the-top gore in The Mean One. *Case 2:* both methods return NO because the candidate (LEGO Marvel Spider-Man) is a superhero action-adventure for older children, whereas the history consists of preschool educational titles (Blippi Wonders, PAW Patrol). *Case 3:* BRIEF shows its advantage. Zero-shot prompting labels The Super Mario Bros. Movie as irrelevant given a history of PG-13 sci-fi/action and a music biopic; in contrast, leveraging collaborative signals, BRIEF correctly assigns YES, as the candidate exhibits high behavioral correlation with Ready Player One and Transformers—capturing shared IP-driven nostalgia not apparent from surface semantics.

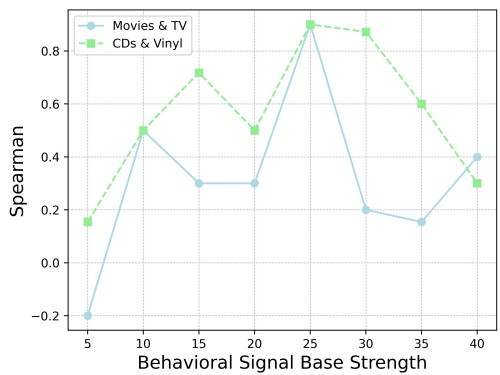

Figure 3: Effect of $\beta$ on reward quality. The plot shows the Spearman correlation between the produced rewards and users' ratings on two datasets (Movies & TV, CDs & Vinyl) as the base strength of the behavioral signal is varied.

### 4 RELATED WORK

**The offline-online gap** The offline-online gap in recommender systems—when offline evaluation fails to predict online performance—arises from many factors, most notably exposure bias: a problem of missing rewards (Jeunen, 2019; Hidasi & Czapp, 2023; Cañamares et al., 2020; Krauth et al., 2020;

Wang et al., 2023; Chen et al., 2019; Rossetti et al., 2016). Off-policy evaluation (OPE) attempts to narrow this gap via methods that estimate how well a new policy would perform using logged data collected under a different historical policy (Swaminathan et al., 2017; Saito & Joachims, 2021; Narita et al., 2021). Inverse propensity scoring (IPS) is a key category of OPE methods: it re-weights observed interactions by propensities to estimate a new model's performance (Mehrotra et al., 2018). Example methods include capped IPS (Gilotte et al., 2018), normalized IPS (Powell & Swann, 1966), normalized capped IPS (Gruson et al., 2019), and self-normalized IPS (Swaminathan & Joachims, 2015; Yang et al., 2018). However, these techniques suffer from high variance (Castells & Moffat, 2022), dependence on accurately estimated propensities (Zhang et al., 2023), and an inability to handle unseen actions (Felicioni et al., 2022). This motivates building reward models to directly impute missing rewards, though these models—trained on biased logged data—can also struggle to extrapolate to items with little or no exposure (Wang et al., 2021).

**User simulators and reward models** User simulators have long been studied for evaluating recommender models: RecLab introduces six hand-crafted simulators (Krauth et al., 2020), RecSim and RecoGym provide configurable reinforcement learning platforms (Ie et al., 2019; Rohde et al., 2018), and Accordion is a trainable Poisson-process simulator (McInerney et al., 2021). Yet these methods suffer from simplified environments, rigid assumptions, or biases in training data. As large language models (LLMs) show remarkable capabilities, they have powered agentic recommenders (Wang et al., 2025a; Zhang et al., 2024b), and researchers are investigating LLM-based user simulators (Yoon et al., 2024): Agent4Rec and SimUSER equip LLM agents with profile and memory modules (Zhang et al., 2024a; Bougie & Watanabe, 2025), while AFL couples a user agent and a recommendation agent in a feedback loop (Cai et al., 2025). These methods still derive rewards from either semantic or behavioral signals alone. The most relevant study to our work employs a collaborative filtering model together with an LLM, but its majority-vote fusion is brittle and heuristic (Zhang et al., 2025b).

**Infusing behavioral signals into LLMs** Research on LLM-based recommenders seeks to overcome the gap between the models' strong semantic understanding and their lack of collaborative knowledge crucial for recommendations. P5 pre-trains a text-to-text model on textualized recommendation data to internalize collaborative patterns (Geng et al., 2022), whereas TALLRec and SOFT fine-tune LLM weights with task-specific interaction logs (Bao et al., 2023; Tang et al., 2025). EAGER-LLM, SeLLa-Rec, and LC-Rec compress behavioral signals into special tokens and fine-tune the LLM to decode them (Hong et al., 2025; Wang et al., 2025b; Zheng et al., 2024). CoLLM and A-LLMRec instead freeze the backbone and learn projectors mapping CF embeddings into the LLM embedding space (Zhang et al., 2025a; Kim et al., 2024). CTRL and LETTER align behavioral and semantic knowledge via contrastive learning (Li et al., 2023; Wang et al., 2024b). These approaches incur heavy training cost and inherit exposure bias from training data. CoRAL avoids training by retrieving user-item interactions, but suffers from long prompts and dependence on high-quality retrieval (Wu et al., 2024). Our proposed training-free method overcomes the aforementioned challenges.

## 5 Conclusion

Large language models are natural candidates for judging recommendation relevance because of their extensive knowledge and strong reasoning capabilities. However, they lack behavioral knowledge that is critical in recommender systems. We introduce BRIEF, a training-free, decoding-time intervention that adaptively biases output logits using conventional collaborative filtering models and, through constrained generation, produces structured evidence for positive rewards. Unlike widely adopted training-based approaches such as Semantic ID, BRIEF operates without finetuning the language model and, to the best of our knowledge, is the first to infuse behavioral signals at decoding time. Future work includes scaling BRIEF to larger-scale reward assignment and combining training-based methods (e.g., Semantic ID) with decoding-time intervention to advance generative recommendation.

## 6 Reproducibility statement

We are committed to ensuring the reproducibility of our research. Our full implementation, including the source code for our models, experiment scripts, and evaluation procedures, is made publicly available in an anonymized GitHub repository at https://github.com/anonymous-submit-code/BrIEF.

The repository contains a detailed README.md file with instructions for setting up the required software environment and executing the code to replicate our results.

## 7 THE USE OF LARGE LANGUAGE MODELS

We utilized the LLM as a general-purpose writing assistant to improve the clarity and polish of the language, which is in line with ICLR policy.

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
