# OpenReview forum: "Behavior-Infused Evidence-First Reasoning: Bridging the Offline-Online Gap in Recommendation"
_ICLR.cc/2026/Conference — ICLR 2026 Conference Withdrawn Submission_

### Official Review · Reviewer_kfUH · 2025-10-15

**Soundness:** 1
**Presentation:** 2
**Contribution:** 2
**Rating:** 2
**Confidence:** 4

**Summary:**

This paper addresses the offline-online performance gap in recommender systems caused by exposure bias. It proposes BRIEF, a training-free method that integrates large language models for semantic reasoning and collaborative filtering (CF) models for behavioral signals. BRIEF enforces evidence-first constrained generation to ensure interpretable rewards and injects behavioral signals via entropy-based adaptive logit biasing during decoding. Evaluations on a mainstream video streaming platform’s A/B tests and public datasets show BRIEF narrows the offline-online gap, improves training data augmentation, and aligns well with real user ratings.

**Strengths:**

BRIEF targets a critical industrial pain point (offline-online inconsistency) with innovative, lightweight design—infusing behavioral signals at the LLM decoding stage (instead of costly fine-tuning) avoids over-reliance on either semantic or behavioral signals alone. The method’s three-module framework (evidence-first generation, behavioral signal extraction, adaptive fusion) is logically coherent, addressing LLM’s post-hoc rationalization and CF’s exposure bias. Experiments are comprehensive: they use both industrial and public datasets, validate across tasks, and ablations.

**Weaknesses:**

### Major Weaknesses
+ Poor MAE-Based Evaluation for Offline-Online GAP (Core Table 2)
  + The offline task scenario (e.g., CTR/CVR prediction vs. staytime estimation) is not fully specified. Please clarify if overlooked.
  + The online metric uses "total streaming hours" (staytime-related), but offline evaluation relies on binary classification (maybe CTR/CVR) instead of staytime estimation—this goal misalignment undermines the method’s validity.
  + Improved streaming hours cannot be solely attributed to solving offline-online inconsistency: online metrics (e.g., impressions, duration) involve trade-offs, and single-metric reporting lacks persuasiveness for isolating BRIEF’s contribution.

### Moderate Weaknesses
+ Insufficient discussion on entropy-based fusion, with no case analyses or dedicated experiments to illustrate its effectiveness.
Inadequate validation of backbone generality—no tests with alternative LLM backbones (beyond Qwen3) or CF backbones (beyond SASRec) to prove the framework’s robustness.
+ Omission of key related work on exposure bias (e.g., ESAM [1]).
+ Lack of trade-off analysis (effect vs. cost) between BRIEF’s "training-free" design and trainable methods (e.g., CoLLM, A-LLMRec).
Unclear differentiation from LLM-based user simulators (e.g., Agent4Rec, SimUSER), with no comparative analysis to highlight BRIEF’s uniqueness.

### Minor Weaknesses
+ Overlooking prior work on "Evidence": Mainstream evidential learning methods (e.g., [2], [3])—which quantify uncertainty and integrate with CF—are not discussed, despite BRIEF’s use of "evidence" for interpretable rewards.
+ Incomplete code repository guidance: No instructions to verify how BRIEF resolves the offline-online gap, only a repository link is provided.
+ Unverifiable offline validation of offline-online inconsistency: This issue depends on online A/B results, and no offline method is proposed to validate BRIEF’s rationality (a field-wide limitation, not a refutable flaw).

[1] ESAM: Discriminative Domain Adaptation with Non-Displayed Items to Improve Long-Tail Performance

[2] Evidential Deep Learning to Quantify Classification Uncertainty

[3] A Comprehensive Survey on Evidential Deep Learning and Its Applications

**Questions:**

Please see Weaknesses

Plus:
BRIEF is designed to fill exposure bias-induced missing rewards but could also serve as a recommender backbone (enhancing collaboration with LLM world knowledge). However, the paper fails to compare BRIEF with existing LLM4Rec works (which also use world knowledge to simulate user feedback) or explain why these works cannot achieve BRIEF’s goals and what makes BRIEF advantageous.

---

### Official Review · Reviewer_iokJ · 2025-10-17

**Soundness:** 3
**Presentation:** 4
**Contribution:** 3
**Rating:** 6
**Confidence:** 2

**Summary:**

This paper tackles a long-standing challenge in recommender systems, the offline–online performance gap, where models that perform well on offline metrics often fail to translate those improvements into real-world gains in online A/B tests. The authors argue that this discrepancy arises primarily from missing or biased feedback in logged data, particularly for items that are never exposed to users. To bridge this gap, the paper introduces BRIEF, a training-free LLM framework that integrates semantic reasoning and behavioral signals for evaluating recommendations. BRIEF enforces a structured, evidence-first generation process, where the LLM must first list evidence items (from the user’s interaction history) before deciding whether a recommendation is relevant. This structure reduces post-hoc rationalization and enforces transparency. Then, at a specific decoding step, BRIEF injects collaborative filtering signals through adaptive logit biasing, modulating the LLM’s token probabilities based on behavioral similarity between items.

**Strengths:**

BRIEF is the first approach to infuse collaborative filtering signals at decoding time rather than through training or embedding alignment. This adaptive logit biasing is conceptually interesting and technically novel, providing a low-cost and flexible way to guide LLM reasoning.

By constraining the LLM to produce explicit evidence before a relevance decision, the paper introduces a reasoning protocol that increases interpretability and reduces spurious correlations, an important step toward transparent AI evaluation.

The decomposition into three modules, Evidence-First Semantic Generation, Latent Behavioral Signal Extraction, and Adaptive Fusion via Logit Biasing, is well-motivated and conceptually clean. The mathematical formulation of adaptive logit biasing, is simple yet effective, capturing the idea that uncertainty should invite stronger behavioral guidance.

**Weaknesses:**

While the paper argues that BRIEF mitigates exposure bias via behavioral fusion, it does not formalize how the combined semantic–behavioral mechanism improves debiasing beyond empirical results. A theoretical discussion of how logit biasing affects bias propagation or variance could strengthen the work.

The behavioral signals rely on a pre-trained CF model (e.g., SASRec). If that model itself suffers from popularity bias or poor calibration, BRIEF may inherit these issues. The paper could provide sensitivity analysis showing how BRIEF’s performance varies with CF quality.

While the paper includes ablations, it lacks analysis of which component contributes most, e.g., what is the relative impact of evidence-first reasoning versus adaptive biasing? A more detailed breakdown would clarify this.

The paper frames BRIEF as generalizable to “structured reasoning tasks with missing feedback,” but it doesn’t demonstrate transfer beyond recommendation domains. Even a small-scale test would make this claim more convincing.

**Questions:**

The results show an inverted-U relation with the base strength parameter β. Would an adaptive or learned β (e.g., dependent on user/item entropy) further improve robustness?

Have the authors examined whether BRIEF reduces or amplifies popularity bias? Since behavioral signals come from existing data, a quantitative bias analysis would be insightful.

---

### Official Review · Reviewer_SWuQ · 2025-10-20

**Soundness:** 3
**Presentation:** 3
**Contribution:** 2
**Rating:** 4
**Confidence:** 4

**Summary:**

This paper proposes a prompting and logit biasing based strategy for LLMs to generate user–item preference scores in recommender systems. The authors conduct both online A/B tests in an industrial production environment and offline experiments on internal and public datasets. The results demonstrate that the proposed approach outperforms compared baselines.

**Strengths:**

1. The paper is generally well-written, with technological details and the methodological framework clearly presented.

2. The authors conduct experiments using both online A/B tests and offline public datasets, demonstrating the method’s potential for real-world applications.

3. The provided code and data enhance the reproducibility of the study.

**Weaknesses:**

1. The novelty and contribution of the paper are rather limited. The work primarily presents an intuitively motivated engineering solution without training to a specific problem within a particular application domain. It offers little conceptual insight or methodological advancement that could generalize to broader research areas. As such, an industrial track would be a better fit for this paper rather than the research track at ICLR.

2. The design of “Evidence-First Semantic Generation” appears superficial and lacks novelty, as similar and more advanced approaches have already been explored in prior work on LLM-based reasoning for recommendation, such as CoT-Rec [1]. The more intersting component, “Adaptive Fusion via Logit Biasing,” is not discussed in sufficient depth. The current exposition remains cursory and seems to rely primarily on intuitive but insufficiently substantiated reasoning.

3. The intuition behind developing a training-free method for industrial recommender systems is not sufficiently convincing. As discussed in related work, many prior studies have introduced collaborative filtering signals to fine-tune LLMs for recommendation tasks. These approaches are more meaningful because they helps bridge the gap between the semantic space of the LLM and the recommendation domain by adapting LLM's inherent knowledge and reasoning ability. In contrast, training-free methods inherently suffer from significant limitations in both representation quality and learning capability. This rationale holds less weight in industrial recommender system contexts, where achieving high recommendation accuracy generally takes precedence over the computational cost of fine-tuning a 32B-parameter LLM.

4. The baselines used in the online A/B tests and on the offline internal dataset are limited to simple, naive methods, excluding more advanced and up-to-date LLM-based approaches such as Agent4Rec. This omission undermines the credibility of the claimed superiority of the proposed strategy in real-world industrial settings.





[1]: Liu, Jiahao, et al. "Improving LLM-powered Recommendations with Personalized Information."

**Questions:**

1. Are there experiments comparing BRIEF with other LLM-based methods in online A/B tests and on offline internal datasets?

2. Has BRIEF been experimentally compared with other fine-tuning-based approaches for incorporating CF signals?

**Details Of Ethics Concerns:**

No ethics concerns.

---

### Official Review · Reviewer_Kqih · 2025-11-02

**Soundness:** 3
**Presentation:** 3
**Contribution:** 2
**Rating:** 4
**Confidence:** 3

**Summary:**

This paper addresses the offline-online gap in implicit-feedback recommender systems caused by exposure bias by proposing a behavior-infused evidence-first reasoning (BRIEF) method. The BRIEF method employs "evidence-first" constrained generation, which requires providing structured evidence before assigning rewards. It also adaptively biases the output token logits using collaborative filtering (CF) signals to infuse behavioral information into the generation process, thereby integrating semantic and behavioral signals. The method's capability for offline evaluation is validated in cross-domain recommendation scenarios where the offline-online gap is particularly significant. Its performance on downstream recommender systems is also verified using benchmark datasets.

**Strengths:**

1.  To address the issue that LLMs excel at semantic understanding but ignore behavioral information, the method injects behavioral signals by adaptively biasing output token logits using collaborative filtering (CF) signals.
2.  BRIEF is a lightweight method that uses "evidence-first" constrained generation, requiring the production of structured evidence before reward assignment to suppress post-hoc rationalization.
3.  It provides a single, predictable control point at the decoding stage for injecting and modulating CF behavioral signals.
4.  It integrates collaborative signals into the LLM's decoding stage rather than the training stage, distinguishing it from methods that rely on training LLMs with special item tokenization or through multi-round conversations. ["Unlike methods that integrate collaborative signals by training LLMs with special item tokenization (Rajput et al., 2023) or through multi-round conversations, BRIEF operates at the decoding stage and is thus training-free."]
5.  The model is not dependent on a specific LLM nor bound to a particular type of CF model, exhibiting a degree of model-agnosticism.

**Weaknesses:**

1.  The claim of being "Training-free" is problematic. While the method does not fine-tune the LLM, it still relies on collaborative filtering/sequential models trained on logged data; thus, the overall approach is not truly training-free.
2.  Directly using behavioral signals computed by the CF model for logit biasing at the decoding control point risks re-injecting exposure bias into the LLM's decision-making process.
3.  The paper does not specify the entity alignment method used in cross-domain recommendation. Furthermore, it lacks quantitative evaluation of alignment accuracy and ablation/robustness experiments analyzing the impact of alignment quality on BRIEF's performance.
4.  There is an inaccuracy in the "4 RELATED WORK" section. The sentence "most notably exposure bias: a problem of missing rewards (Jeunen, 2019; Hidasi & Czapp, 2023; Cañamares et al., 2020; Krauth et al., 2020; Wang et al., 2023; Chen et al., 2019; Rossetti et al., 2016)" describes exposure bias as a problem of missing rewards, whereas the preceding context in the introduction characterizes it more broadly as missing feedback.

**Questions:**

1.  Could the authors systematically compare their method against mainstream Off-Policy Evaluation (OPE) baselines (e.g., IPS, SNIPS, DR/Switch-DR, SLOPE, DM, DM+IPS) to validate its effectiveness in narrowing the offline-online gap?
2.  Does using [ or [] as the YEA/NAY trigger, which strongly depends on the tokenizer and formatting (spaces, line breaks, Markdown), make it prone to mismatches or false triggers when switching models?
3.  It is recommended that the authors provide pseudo-code for the BRIEF algorithm to clarify the specific logic of its implementation.

---

### Note · Authors · 2025-12-09

I have read and agree with the venue's withdrawal policy on behalf of myself and my co-authors.